# “Resistance Is Futile”: A Pilot Study into Pseudoresistance in Canine Epilepsy

**DOI:** 10.3390/ani13193125

**Published:** 2023-10-06

**Authors:** Filip Kajin, Nina Meyerhoff, Marios Charalambous, Holger Andreas Volk

**Affiliations:** 1Department of Small Animal Medicine and Surgery, University of Veterinary Medicine, 30559 Hannover, Germany; fkajin@vef.hr (F.K.); nina.meyerhoff@tiho-hannover.de (N.M.); marios.charalambous@tiho-hannover.de (M.C.); 2Clinic for Internal Diseases, Faculty of Veterinary Medicine, University of Zagreb, 10000 Zagreb, Croatia

**Keywords:** seizures, dog, pseudoresistance, refractory epilepsy

## Abstract

**Simple Summary:**

We introduce the term and evaluate the phenomenon of pseudoresistance to antiseizure-medication in canine epilepsy cases initially diagnosed with medication-resistant epilepsy. Our study shows that around one third of the patients initially diagnosed with medication-resistant epilepsy had in fact other underlying reasons for their drug resistance (they were in fact pseudoresistant) and all but one of these cases responded to modification of the initial therapy protocol.

**Abstract:**

Epilepsy is a common neurological disorder in veterinary practice, complicated by frequent occurrence of medication-resistant epilepsy. In human medicine, it has been noted that some patients with medication-resistant epilepsy have in fact other reasons for their apparent medication-resistance. The aim of this retrospective study was to describe the issue of pseudoresistance using as an example a population of dogs presented with presumed medication-resistant epilepsy and provide an in-depth review of what is known in human medicine about pseudoresistant epilepsy. One-hundred fifty-two cases were identified with medication-resistant epilepsy, of which 73% had true medication-resistant epilepsy and 27% patients had pseudoresistance. Low serum anti-seizure medication levels were the most common cause of pseudoresistance, present in almost half of the cases (42%), followed by inadequate choice of drugs or dosages (22%), misclassification (22%) or misdiagnosis (9%) of epilepsy and poor compliance (9%). All cases of pseudoresistance, except for one, responded to a modification of the initial therapy protocol. Pseudoresistance can bias clinical trials, misinform the clinical decision-making process, delay diagnosis and treatment, and misinform owners about their pets’ prognosis. A substantial proportion of these cases can have improvement of their seizure frequency or achieve seizure freedom upon modification of their therapeutic protocol.

## 1. Introduction

Epilepsy is a common neurological disorder in veterinary practice, complicated by frequent occurrence of medication-resistant epilepsy [1,2,3,4]. Medication-resistant epilepsy (MRE) is defined as failure of adequate trials of two tolerated, appropriately chosen antiseizure medication (ASM) schedules (whether as monotherapies or in combination) to achieve sustained seizure freedom [4,5] or achieve patient-specific therapeutic success [6]. Recently, it was agreed that partial therapeutic success (i.e., reduction of seizure frequency and severity compared to baseline and prevention of status epilepticus and cluster seizures, rather than just seizure freedom) is an essential goal in veterinary patients [6]. More than two thirds of dogs with epilepsy will continuously experience seizures long-term and around one third will be classified as medication-resistant despite adequate treatment with at least two ASM, such as phenobarbital (PB) and/or potassium bromide (KBr) [3]. Early recognition of MRE is important to assure that patients can receive possible different medical or alternative therapeutic approaches [6,7].

In human medicine, it has been noted that not all patients with recurrent seizures or epilepsy, despite prescription of appropriate ASMs, have MRE [8,9]. Consequently, pseudoresistance or “false pharmacoresistance” has been proposed as an umbrella term, comprising cases with lack of adherence to prescribed ASM, wrong classification of epilepsy, misdiagnosis and inadequate choice of drugs or dosage [7,8,10]. In the United Kingdom (UK), Booth et al., already described subpar owner compliance in administrating ASM to dogs with epilepsy [11]. Only one in five owners were 100% compliant. In addition, up to one in ten dogs presenting with seizures have reactive seizures rather than epilepsy [12]. This, along with other reasons for pseudoresistance could represent a significant and poorly described issue in management of epilepsy in veterinary medicine.

The aim of the current study was, therefore, to describe the issue of pseudoresistance using as an example a population of dogs presented with presumed MRE to our hospital and provide the reader with an in-depth review of what is known in human medicine about pseudoresistant epilepsy.

## 2. Materials and Methods

### 2.1. Patients

This was a single-center, retrospective study conducted by searching electronic records of dogs presenting to the Department of Small Animal Medicine and Surgery of the University of Veterinary Medicine, Hannover, between 2015 and 2022 with presumably uncontrolled epileptic seizures. Cases were searched for following key words in German “seizure”, “resistant”, “refractory”, “epilepsy”, “poorly controlled “and “idiopathic epilepsy”.

### 2.2. Definition Parameters

Medication-resistant idiopathic epilepsy was defined as previously described [4,5], i.e., only cases of idiopathic epilepsy with failure to achieve seizure freedom with ≥2 ASMs. Serum concentrations within the therapeutic range (PB > 15 mg/L, KBr > 1000 mg/L) and/or treatment with adequate dosages (imepitoin 10–30 mg/kg BID; levetiracetam (LEV) 20 mg/kg TID) were needed to be documented to classify as MRE [13].

Cases of dogs with pseudoresistance were defined as and classified into the following categories in adherence to literature in people: “misdiagnosed” (not epileptic seizures or presence of reactive seizures), “misclassified” (seizures due to a structural brain lesion), “low serum ASM levels” (documentation of serum ASM concentrations below the therapeutic range), “inadequate choice of drugs or dosage” (treatment with subtherapeutic dosages or with drugs deemed inadequate based on the International Veterinary Epilepsy Task Force (IVETF) [13]), and “poor compliance” (patient owners non-adhering to the drug application plan) [7,8,10,14,15]. Different causes for pseudoresistance are depicted in Figure 1, adapted from the research of Bašić [10] and modified to better fit veterinary medicine.

For each patient with pseudoresistance, the signalment, onset of the first seizure (in months), age at presentation (in months), final diagnosis (if not idiopathic epilepsy) was noted. Follow-up was defined as short term (less than 12 months) or long term (12 months or longer). Seizure freedom (>3 times longest pretreatment interictal interval and at least 3 months) and partial improvement (prevention of cluster seizures or status epilepticus, relevant reduction of seizure frequency considering pretreatment seizure frequency or reduction in seizure severity) were defined as previously described [4]. Response to the change of the therapeutic plan was defined as “response” (seizure freedom or partial improvement) or “no response”. The cumulative drugs given as long-term therapy were also noted.

### 2.3. Data Management and Analysis

Data was collected and analyzed using Microsoft^®^ Excel for Mac (Version 16.73).

## 3. Results

Overall, 1872 cases were retrieved after using the initial key words. Figure 2 shows a flow-diagram of the exclusion process. Most of these cases were excluded due to not classifying as medication-resistant epilepsy patients. Cases were also excluded due to incomplete medical records, drug-naïve patients and patients receiving only one ASM. Patients referred with MRE were included even if they received only one ASM, and hence, they were classified into the “inadequate drug or dosage” category. Following review of the medical records, 152 cases were identified with MRE diagnosed in-house or referred from other veterinarians as such. One hundred eleven (73%) of the patients with the initial diagnosis of MRE were determined to have true MRE and 41 (27%) patients were classified as pseudoresistant.

The most prevalent breed of dogs with pseudoresistance in our study were mix-breed dogs (10/41), followed by Labrador retriever (4/41), French bulldog (3/41), Australian Shepherd (2/41), Boxer (2/41), Husky (2/41) and one of each; Airedale Terrier, Bolonka Zwetna, Border Collie, Briard, Dachshund, German Sheep Poodle, German Shorthaired Pointer, Jack Russel Terrier, Landseer, Maltese dog, Mexican hairless dog, Miniature schnauzer, Standard poodle, Rottweiler, Rough collie, Sheepdog, Toy poodle, Vizsla. Sixteen patients were female (five entire and 11 neutered) and 25 were male (13 entire and 12 neutered). The data about the age of first seizure and age at presentation was available for all dogs in this study. The median age of onset of the first seizure was 15.5 months (range: 9–25, IQR: 9.5) and the median age at presentation was 50.5 months (range: 10–160, IQR: 64).

Phenobarbital was the most common ASM administered to 38 patients in our study, followed by imepitoin (n = 23), KBr (n = 18), levetiracetam either as chronic or pulse therapy (n = 16), gabapentin (n = 3), and clonazepam as pulse therapy (n = 2). Three patients received only one ASM, 21 received two, 13 received three, and four patients received four ASM.

Most of the patients with pseudoresistance had low serum ASM levels (16/41), followed by misclassified patients (9/41), inadequately chosen drugs or dosages (7/41), misdiagnosed patients (3/41), poor owner compliance (2/41), inadequately chosen drugs/dosages and low serum ASM levels (2/41), misdiagnosed and low serum ASM levels (1/41), and misclassified and inadequate choice of drugs/dosages (1/41). The distribution of patients from different main pseudoresistance categories is shown in Figure 3.

Of the patients with low serum ASM levels, 11 had a low serum PB level (median 12 mg/L, range 5–14.8 mg/L), eight patients had a low serum KBr level (median: 762 mg/L, range: 533–900 mg/L) and one patient had low levels of both PB (12.2 mg/L) and KBr (724 mg/L). Patients that had structural epilepsy and were misclassified as having MRE had the following diagnoses: a focal intra-axial mass lesion suspected to be of neoplastic origin (n = 3), post-traumatic epilepsy (n = 2), suspected hippocampal sclerosis/necrosis (n = 1), a dog that initially had IE and later developed structural epilepsy (diagnosed based on IVETF guidelines, no further diagnostics were permitted by the owner) (n = 1), meningoencephalitis of unknown origin (MUO) (n = 1), partial holoprosencephaly (n = 1) and Lafora disease (n = 1). Patients with inadequately chosen drugs or dosages received imepitoin for cluster seizure activity and/or status epilepticus (n = 5), received a combination of PB and levetiracetam (n = 2), received a combination of imepitoin and levetiracetam (n = 1), a subtherapeutic dose of PB (n = 1), and a subtherapeutic dose of PB and imepitoin (n = 1). Misdiagnosed patients had the following diseases which were misdiagnosed as MRE: insulin-producing neoplasia (n = 2), initially IE and later developed hypercalcemia due to parathyroid neoplasia (n = 1), and paroxysmal dyskinesia (n = 1). Poor owner compliance was due to patient owners missing dose administrations (n = 1) or reducing doses of ASM (n = 1) without consulting the veterinarian. Appendix A shows the distribution of patients within subcategories of different pseudoresistance categories.

Median follow-up time was 15 months (range: 0–144 months, IQR: 41). Short term follow up was available in 20/41 patients and long term in 22/41 patients. Data on therapy response was available for 33 cases. Fifteen cases within the “low ASM category achieved a response after a change of the therapeutic plan (increasing the dose of ASM). Five patients from the “misclassified” category achieved a therapeutic response to a change in the ASM plan or adjunctive therapy for their underling condition, and one case did not. Follow up was available for the following patients: post-traumatic epilepsy (2), intracranial neoplasia (1), Lafora disease (1), and the dog that initially had IE and later developed structural epilepsy (1) all of which achieved a therapeutic response, while the patient with the partial holoprosencephaly (1) failed to achieve a therapeutic response. Six patients from the “inadequate drug/dosage” category achieved therapeutic success to a change in the ASM plan. Four patients from the “misdiagnosed” category achieved therapeutic success to a change of therapy. Finally, from the “poor owner compliance” category, both patients achieved therapeutic success. Appendix A displays an overview of the response in different pseudoresistance categories. Three patients were euthanized because of epilepsy and four other patients died or were euthanized because of unrelated reasons. Seven patients were lost to follow up upon or soon after being diagnosed with pseudoresistant epilepsy.

## 4. Discussion

Medication-resistance remains one of the major challenges in veterinary epileptology, as the majority of dogs diagnosed with idiopathic epilepsy continue to have seizures despite ASM [3,4,16]. However, a proportion of dogs referred with a MRE diagnosis do indeed not have MRE, which can bias clinical trials, and misinform the clinical decision-making process, delay diagnosis and treatment, and misinform owners about their pets’ prognosis. We therefore introduce here the term pseudoresistant epilepsy in veterinary epileptology. In human medicine, several groups have recently reported that many patients with uncontrolled epilepsy do not have refractory epilepsy, but suffer from pseudoresistant epilepsy, and can indeed attain seizure freedom [9]. In the current study a third of the dogs initially thought to have MRE were diagnosed with pseudoresistant epilepsy, with two thirds of those having inadequate treatment and the other third being misdiagnosed or misclassified. This highlights the importance of improving veterinary education, as priorly indicated by interest groups [17].

The IVETF published multiple consensus reports in 2015, which served to “set out a unified and standardized set of guidelines for the research, diagnosis and treatment of canine and feline epilepsy for the first time ever in veterinary medicine” [4,13,18]. Many of the classifications, definitions and terminology lean on the one established by International League Against Epilepsy (ILAE) [5]. This is also the case for the definition of medication-resistant epilepsy, which is defined as failure of adequate trials of two well-tolerated, appropriately chosen and used ASM schedules (whether as monotherapies or in combination) to achieve sustained seizure freedom [4,5]. Our study shows that around one third of the patients diagnosed prior to referral with MRE had in fact other underlying reasons for their seemingly MRE, and were in fact pseudoresistant. It seems to be a more prevalent issue than previously considered; a fact also being recognized in people. Hao et al., described pseudoresistance ten years ago in a study in people undertaken on a newly recognized and chronic cohort of patients with uncontrolled epilepsy, where 56% (newly recognized) and 41% (chronic cohort) did not meet the ILAE proposed definition of true MRE [9]. The same year, a different group also investigated various possible reasons for uncontrolled seizures to determine the different forms and impact of what they named pseudointractability [14]. In their study, 60% of the patients with uncontrolled seizures had in fact pseudoresistant epilepsy [14].

In the current study, we observed that the most prevalent reason of pseudoresistance were low serum ASM levels in patients receiving adequate dosages, which we observed in almost half of the patients with pseudoresistance. Serum ASM levels should be monitored regularly, as previously defined [13,19]. Low serum concentrations are associated with treatment-failure, and excessively high ones with toxicity [13,20]. Possible reasons for low serum ASM levels can be poor owner compliance/adherence to the treatment plan or re-check appointments, therapeutic failure due to metabolic tolerance (like described for levetiracetam), or suboptimal absorption of ASM from the gastrointestinal tract [11,13,21,22,23,24]. A special focus should be to develop strategies to improve owner compliance. A recent study showed that only one in five owners had a 100% compliance giving ASM correctly [11]. The overall median compliance was only 56%. These factors underline the importance of improving the training of veterinary surgeons and owners alike of the importance to adhere to treatment protocols [17].

The second most common reasons for pseudoresistant epilepsy in the study were inadequately chosen ASM (or dosages of ASM) and misclassified epilepsy. This is in concurrence with human medicine, where a substantial proportion of patients with “uncontrolled” epilepsy received the wrong ASM or a suboptimal dosage of an appropriate ASM [7,8,14,15]. In veterinary clinical practice in Europe, legal regulations define the use of PB, KBr, and imepitoin as first-line ASM in dogs with idiopathic epilepsy [13,25]. We considered LEV as sole chronic add-on treatment to PB, KBr or imepitoin to be inadequate, due to its known “honeymoon effect” and questionable efficacy compared to placebo or PB [16,22,24,26]. Also, since the efficacy of imepitoin has not yet been proven in dogs with cluster seizures or status epilepticus, we considered its use for this purpose as inadequately chosen ASM [13]. Treatment failure is not uncommon in canine idiopathic epilepsy, still, patients with structural epilepsy are more prone to develop drug resistance and have shorter survival times [27,28]. For this reason, we believe that the term “medication-resistant epilepsy” be reserved for patients diagnosed with idiopathic epilepsy, since drug resistance is a phenomenon to be expected to some extent in patients with structural epilepsy, especially if the main lesion is not diagnosed, misdiagnosed or the patient is not receiving therapy (apart from ASM) for the underlying disease.

Misdiagnosis of epilepsy was seen in 9% of the pseudoresistant patients in this study, reflecting also what is seen in human medicine, where patients with psychogenic nonepileptic seizures, vasovagal syncope, cardiac arrhythmia, sleep/movement-disorders, or psychiatric disorders are misdiagnosed as having epilepsy, which later fails to improve on ASM and becomes classified as MRE [8,9]. In our study, two dogs referred as medication-resistant epilepsy cases, were found to have hypoglycemia (suspected due to insulin-secreting neoplasia) as the cause for the recurrent seizures, both of which had a short-term improvement after discontinuing the ASM, and addressing hypoglycaemia with dietary and medical therapy, before being lost to follow up. Reactive seizures can sometimes be misdiagnosed as epilepsy cases, and studies show that up to 11% of patients presenting with seizures can suffer from reactive seizures due to metabolic or toxic causes [12]. Another misdiagnosed case from this study is that of a dog with probable paroxysmal dyskinesia, which responded to a gluten-free diet. Paroxysmal dyskinesias are being more and more frequently recognized as seizure look-alikes, and some forms are associated with gluten sensitivity and respond well to a gluten-free diet but often not to commonly used ASM [29]. Two of the patients in our study were initially diagnosed with idiopathic epilepsy (IVETF Tier II confidence level), and treated successfully for a long period of time and later developed additional conditions which led to poor seizure control. One developed hypercalcemia (suspected due to parathyroid neoplasia) and one developed structural epilepsy. Hypercalcemia can be associated with seizures, either due to reduced neuronal membrane excitability, or hypercalcemia-induced hypertensive encephalopathy and vasoconstriction [30]. This highlights the need for regular and thorough re-checks in patients chronically receiving ASM for IE, as they can in future develop other seizure-inducing conditions which can disrupt their previously established seizure control.

Finally, only 5% of patients in this study were found to be pseudoresistant due to poor owner compliance. This is different to the aforementioned UK study investigating ASM compliance [11]. The authors found a significant association between polytherapy and compliance, indicating that patients which receive multiple ASMs are more likely to be compliant. This is a possible explanation to why this group of pseudoresistant cases was seen less frequently than other ones, as they already mostly received two or more ASM. Furthermore, drug compliance usually improves prior to presentation to a doctor or could be one of the reasons why so many dogs had subpar ASM serum levels in the current study [31]. Owners might not readily give the information that they do not give adequately the medication.

A notable finding in our study was that all cases, except for one, responded to a modification of the initial therapy protocol. The one case that did not respond was of a misclassified dog which had structural epilepsy and failed to achieve seizure freedom or at least a partial response. As discussed above, medication-resistance is more commonly seen in patients with structural epilepsy if the underlying brain disease cannot be addressed [24,25]. This calls attention to the importance of a thorough approach to cases with presumed MRE to identify pseuodresistance, as seizure control can be dramatically improved in many of these patients and, consequently, their quality of life upon correction of the therapeutic protocol. Seizure freedom is the main aim in therapy of epilepsy, but MRE continues to be a major clinical issue in the therapeutic management with substantial implications for quality of life and survival times [28]. The importance of striving for seizure freedom has been highlighted in a recent retrospective study of probable sudden unexpected death in epilepsy (pSUDEP), which highlighted that any seizure can potentially lead to a fatal outcome [32]. Also, uncontrolled recurring epileptic seizures are correlated with increased stress levels (for both dogs with epilepsy and their owners), a poor quality of life, and a limited therapeutic response, all of which can result in the decision for earlier euthanasia [3,28,33,34,35]. In a recent human and veterinary medicine collaborative research, it was agreed that clinicians should recognize and treat not only the epileptic seizures, but also any behavioral comorbidities, especially in patients with MRE [6]. Limiting trigger factors, such as stress in both dogs and owners is essential as this could substantially improve seizure control and both canine and human quality of life [6].

Misdiagnosis of epilepsy is a common problem in human medicine, and carries with it all of the secondary handicaps and limitations of epilepsy: the stigma and social marginalisation, lifestyle limitation, employment and driving restrictions, side effects and potential teratogenic effects of ASM [36]. Between 13 and 42% of human patients with epilepsy are misdiagnosed, which highlights the importance of this issue in human epileptology [37]. A significant portion of individuals diagnosed with MRE experience psychogenic non-epileptic seizures. Among patients admitted to video-EEG monitoring units with a MRE diagnosis, it is discovered later that one in every four to five patients actually suffers from non-epileptic events, with the majority being of psychogenic origin [15]. Psychogenic non-epileptic seizures manifest as sudden motor, non-motor, or behavioral changes resembling epileptic seizures but lack EEG (electroencephalogram) correlates. These conditions are typically attributed to responses to distress or behavioral issues [38]. Dogs may display behavioral conditions that bear similarities to specific human psychiatric disorders, including separation anxiety, obsessive-compulsive disorder, cognitive dysfunction, dominance aggression, and panic disorder. These canine conditions exhibit notable correlations with their human counterparts, such as generalized anxiety disorder, obsessive–compulsive disorder, Alzheimer’s disease, impulse control disorders, and panic disorder [39]. However, psychogenic non-epileptic seizures have currently not been described in veterinary medicine, and although relatively common in humans, do not seem to be a cause for pseudoresistance in canine patients. Occurrences such as syncope or psychogenic pseudo-syncope can be confused with epilepsy in humans due to their overlapping clinical characteristics. These shared features include sudden loss of consciousness without prior warning, unusual limb movements like myoclonic jerks or tonic-clonic activity, and episodes of incontinence [37]. Petkar et al., found a high incidence of bradyarrhythmias and asystole in cases where there was uncertainty surrounding the diagnosis of epilepsy [37]. Cardiac channelopathies often manifest with convulsive syncope, leading to a misdiagnosis of epilepsy which puts these patients at risk of developing serious and life-threatening arrhythmias [40]. Differentiation of generalised epileptic seizures and syncope can also be quite challenging in veterinary medicine, since the two syndromes can share a similar clinical manifestation [41,42]. None of the dogs in this study had an arrythmia misdiagnosed as MRE as a cause for pseudoresistance. This could reflect the fact that epilepsy, rather than syncope, was correctly recognised as a cause of episodic loss of consciousness or abnormal limb movements for the animals in this study. It is, however, possible, that a lot of these remain underdiagnosed or misdiagnosed, so future studies should aim to see if convulsive syncope is a more widespread phenomenon in veterinary medicine than previously reported.

The current study has significant limitations mainly due to its retrospective nature. It is likely that many cases with pseudoresistance were missed when clinical records were searched for, as they were not yet classified as such. Hence also the distribution of pseudoresistance categories needs to be interpreted with caution. Nevertheless, they do echo what has been found in human medicine. We would propose that the term “pseudoresistance” and the different categories are used in clinical records, so that this can be better evaluated in future studies. This will also help to address better training of owners and veterinary surgeons alike.

Also, the small number of patients and different pseudoresistance categories, which include a fairly inconsistent distribution of patients with different etiopathologies, make any statistical analysis unfeasible. Further studies could aim at investigating reasons for pseudoresistance in more detail, with an aim to use statistical methods to provide a better idea on the prognostic factors. This study included only canine patients with pseudoresistance. Similar to dogs and humans, most feline patients with epilepsy respond to ASM therapy and around one third of cases do not which results in a poor outcome and quality of life [43,44]. Further studies are warranted to describe the phenomenon of pseudoresistance in this species.

## 5. Conclusions

In conclusion, this is the first study to introduce the term and evaluate the phenomenon of pseudoresistance in epilepsy in veterinary medicine. The results showed that a substantial proportion of these cases can have short- or long-term improvement of their seizure frequency, or achieve seizure freedom upon modification of their therapeutic protocol.

## Figures and Tables

**Figure 1 animals-13-03125-f001:**
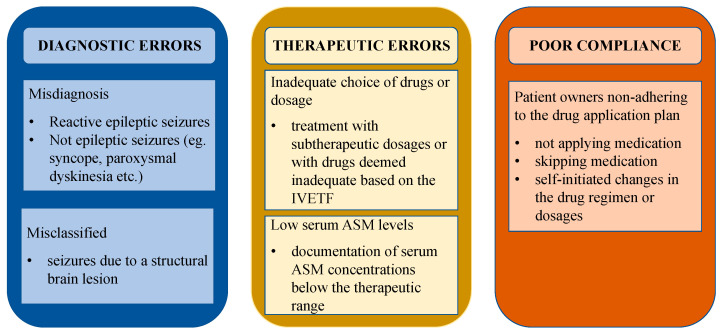
Causes of pseudoresistance. Different causes for pseudoresistance adapted from human medicine and modified to better fit veterinary medicine. Abbreviations: IVETF—International Veterinary Epilepsy Task Force, ASM—antiseizure medication.

**Figure 2 animals-13-03125-f002:**
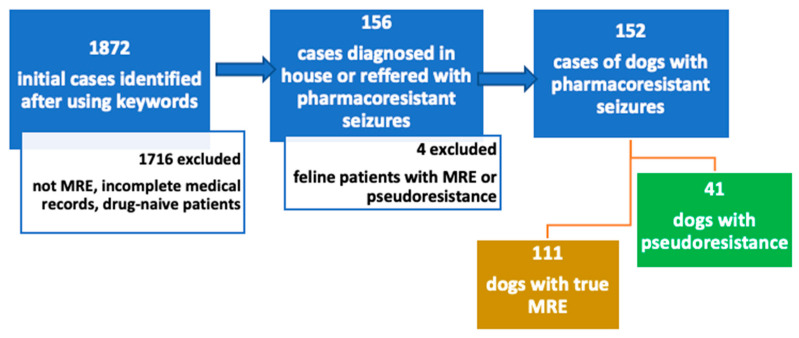
Flow-chart of inclusion and exclusion criteria. Cases were retrospectively searched between 2015 and 2022 using following key words in German “seizure”, “resistant”, “refractory”, “epilepsy”, “poorly controlled “and “idiopathic epilepsy”. Abbreviations: MRE—medication-resistant epilepsy.

**Figure 3 animals-13-03125-f003:**
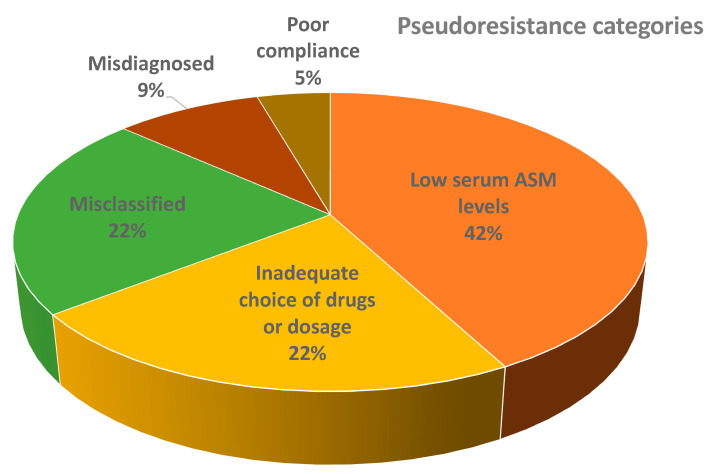
Distribution of patients from main different pseudoresistance categories (the figure does not show combinations of groups). Abbreviations: ASM = anti-seizure medication.

## Data Availability

Not applicable.

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
