# Peer review of "“Resistance Is Futile”: A Pilot Study into Pseudoresistance in Canine Epilepsy"

_animals, 2023, doi:10.3390/ani13193125_

Round 1
Reviewer 1 Report
General comment
This study introduces the concept of and defines “pseudoresistance” in veterinary practice. The reviewer has also pondered the importance of pseudoresistance in canine and feline epilepsy, so generally welcomes this study and the authors’ opinion. However, in this version of the manuscript, there are a few minor corrections and discussable points listed below. The reviewer will accept this study after reasonable revision.
Specific comments
1. Line 10 and throughout the manuscript: As antiepileptic/antiseizure ‘drug’ (AED/ASD) has been changed into antiseizure ‘medication’ (ASM) in recent years, the term “drug-resistant epilepsy (DRE)” is in chaotic state. The reviewer sometimes sees “medication-resistant epilepsy (MRE)” or “ASM-resistant epilepsy”. ILAE does not comment on this yet, so we could not which is preferred. This is just my comment, but I want to know the authors’ opinion. And, if you agree with my idea, this reviewer recommends inserting “medication-resistant epilepsy (MRE)” after the first appearance of “drug-resistant epilepsy” (Line 33).
2. Line 13: “responded to…protocol.” Please correct the Font.
3. Line 43 and throughout the manuscript, including Supplementals: The abbreviation of potassium bromide. In this manuscript, the authors used “PBr”. However, all major papers use “KBr” for potassium bromide. And PBr is very confusable to PB. Please replace all PBr with KBr.
4. Line 70: Please define “seizure freedom” for this study. The authors describe about “partial therapeutic success” in the Introduction (lines 36-40), and the reviewer thought they intended to include ‘partial therapeutic success’ in ‘seizure freedom’ here. If so, here should be changed to “essential goal” or define “seizure freedom” of this study.
5. Line 76 and elsewhere: The authors used “reactive epileptic seizures” in this paper, but IVETF defined “reactive seizures”, not inserted ‘epileptic’.
6. Line 106: Perhaps ‘drug-resistant seizures’ after DRE in this sentence should be deleted.
7. Lines 113-119, especially 118-119, and Fig 2: This is canine study and excludes feline patients (Figure 2), but in Lines 118-119, the authors state “The one cat was a domestic short hair cat.” What’s this? And if excluding this 1 cat, the number of dogs with pseudoresistance will be 40. Please confirm and correct all data.
8. Line 125: potassium bromide –> KBr
9. Line 140: 12,2 –> 12.2
10. Lines 148-150: I disagree that “PB+LEV” and “IMP+LEV” are “inadequately chosen drugs”. Why those combinations are inadequate? Is it because it is not written in IVETF/ACVIM recommendation?
11. Line 148-149: phenobarbital –> PB
12. Line 192: antiepileptic drug –> ASM
13. Line 222: In clinical practice –> In veterinary clinical practice
14. Line 251: Regarding “IE (Tier 2)”. I think the authors did not abbreviate idiopathic epilepsy as “IE” before that, and non-neuro readers might not understand the meaning of “Tier 2”.
15. Figure 4S: Please change PBr into KBr. Not ‘hipocampal’, the “hippocampal”. Should abbreviate imepitoin as IMP (not Imep). And abbreviations KBr, IMP, and MUO must be opened in the legend.
Reviewer 2 Report
This is a very well-written article adding valuable information to the epilepsy literature. The quality of the written language is excellent with minimal text editing required. A few examples are highlighted below along with some specific comments on the contents.
A few minor text editing questions/checks for example:
Line 82: the reference says 'Basic et al. (10) but the reference list is for a single author
Line 116: apparent punctuation missing 'German Sheep Poodle German Shorthaired Pointer,)
Line 240 and Line 252 the use of the abbreviation susp. should be reconsidered.
COMMENT 1
At several points in the manuscript, the definition of drug-resistant epilepsy is described as failure of adequate trials of two tolerated, appropriately chosen anti-seizure medication schedules (whether as monotherapies or in combination) to achieve sustained seizure freedom or achieve patient-specific therapeutic success.
It might be helpful to the reader to have a brief supplementary comment on the definition used in this and in other veterinary studies, specifically the use of '≥2 anti-seizure medications' versus the consideration for the medication being either as ‘monotherapy or in combination’.
The definition as in line 70 that states 'failure to achieve seizure freedom' versus 'failure to achieve seizure freedom or achieve patient-specific therapeutic success'. It might be helpful to comment on the choice around using seizure-free versus partial therapeutic success in such studies etc. Whilst there is comment on the importance of seizure freedom, the likelihood of ongoing seizures versus seizure freedom in canine patients raises questions about whether the absence of seizure freedom should be in the definition of drug resistance.
COMMENT 2
Line 113: 'the most prevalent breed of dog in our study were mix-breed dogs (10/41)..... ‘
It may be easier for the reader to follow if there is clarity on whether this is referring to the dogs with pseudoresistance (41 dogs) versus all dogs with drug-resistant epilepsy etc (152 dog), or all cases diagnosed with drug resistance epilepsy (156) etc. There is a feeling of confusion when reading the breed list (41 dogs), with the following statement being 'The one cat was a domestic short hair cat'.
COMMENT 3
Line 151: this line starts the discussion on misdiagnosis, and specifically concerning the patient initially diagnosed with idiopathic epilepsy and later developed hypercalcemia due to parathyroid neoplasia.
Is the hypothesis that this patient's response to their anti-seizure medication schedule became poor, for example, due to the onset of hypercalcemia secondary to parathyroid neoplasia? Or due to another paraneoplastic phenomenon? It is not immediately clear to the reader since seizures are not the typical neurological manifestation of hypercalcemia, especially not in isolation, although of course mechanisms including hypercalcemia-induced cerebral vasoconstriction do exist. It might be helpful to add some extra commentary on this to improve the understanding of the category of misdiagnosis in cases of peusdoresistance.
COMMENT 4
Line 219: This sentence reads 'these two groups make up more than the other half of patients with pseudoresistant epilepsy' referring to inadequately chosen ASM (or dosage of ASM) and misclassified epilepsy. This sentence is confusing to the reader.
Low ASM was noted to be the most prevalent reason for psudoresistance at 42% versus 58% being another cause.
Inadequately chosen ASM being 22% and misclassified being 22%.
It is challenging to follow whether the sentence in line 219 means that these two groups combined (total 44%) make up 'more than the other half of patients with pseudoresistant epilepsy' (ie more than half of the 58% which are not low ASM?). You might consider restructuring this sentence/discussion to avoid any confusion.
COMMENT 5
Considering 'misclassified' as a category of pseudoresistance, 5 of 6 for which there was follow-up, were classed as having a response to changes in the therapeutic protocols, 'a change in the ASM plan or adjunctive therapy for their underlying condition'.
We are not aware of which pathology these 6 cases with follow-up were diagnosed with, out of 3 x intracranial neoplasia, 2 x post-traumatic epilepsy, 1 x hippocampal sclerosis/necrosis, 1 x undefined, 1 x MUO, 1 x anomaly, 1 x lafora disease.
There are potentially 6 disorders that could have adjunctive therapy for their underlying condition that is not an anti-seizure mediation (eg. intra-cranial neoplasia (3), MUO (1), anomaly (1), undefined (1)) but we are potentially left with the interpretation that some of the treatment and reported response to changes in the therapeutic protocol is simply ASM modification.
We are also left to assume that the ASM schedule for these patients already included adequate drug choices with serum concentrations within the therapeutic range, but that modification of the change in ASM plan involved increasing drug dosages i.e., raising the serum concentrations higher within the therapeutic range.
This all leads to an as yet unaddressed question of whether it is enough to assess and categorize the drug response of these patients, based on the definition parameter 'documentation of serum concentrations within therapeutic range' when the low end of the therapeutic range for phenobarbitone (>15mg/L) and bromide (>1000mg/L) still leaves a large amount of scope for dosage increase.
Is a dog really drug-resistant if the seizures are uncontrolled with phenobarbitone serum level 15mg/L and bromide level 1000mg/L, but then controlled at phenobarbitone serum levels 25mg/L and bromide level 1800mg/L?
You may consider adding a discussion point to cover this as an example of the complicating factors when defining drug-resistance etc.
This is a well-written article with minimal changes needed and a good quality of written English language.
Reviewer 3 Report
Dear AA. thank you for submitting this paper that would like to assess presence of pseudoresitency in dogs. The paper sounds. I would change the title since the "residence is futile" it is not appropriate in this context.
Reviewer 4 Report
The authors introduce the term, describe and analyze the prevalence of pseudoresistance in canine epilepsy - a very interesting and debatable notion even in human medicine. The paper is well written and I believe the topic might be relevant for the journal readers.
General remarks:
Based on the results of a retrospective study, the authors show the pseudoresistance (as is defined in humans medicine) is present in epileptic dogs. However, the results show that if the IVETF recommendation are followed, up to 95/100 dogs will respond to ASM. Thus, the introduction of pseudoresistance in veterinary medicine may be dangerous: will the inability to follow the IVETF recommendations be masked/attenuated by this new term? I recommend the authors to better underline the need of a sustained effort to educate both the owner and vets.
Specific remarks:
Results – It is possible to quantify how many dogs where initially diagnosed with Tier I - II or more/ initially diagnosed by a specialist or a general practice vet? I believe the aspect may be relevant in understanding/identifying the critical points involved in the occurrence of such a large proportion of dogs with low ASM (see line 330).
Line 222 – “legal regulation…” reference needed
Line 225 – “We considered LEV ………… due to its known “honeymoon effect”. –reference needed to better clarify if the statement is referring to idiopathic epilepsy dogs in general or only to pharmacoresistant ones.
“and questionable efficacy compared to placebo or PB”. Reference needed.
